# Green Synthesis of Silver Nanoparticles Using *Paullinia cupana* Kunth Leaf Extract Collected in Different Seasons: Biological Studies and Catalytic Properties

**DOI:** 10.3390/pharmaceutics17030356

**Published:** 2025-03-10

**Authors:** Alan Kelbis Oliveira Lima, Ítalo Rennan Sousa Vieira, Lucas Marcelino dos Santos Souza, Isadora Florêncio, Ingrid Gracielle Martins da Silva, Alberto Gomes Tavares Junior, Yasmin Alves Aires Machado, Lucas Carvalho dos Santos, Paulo Sérgio Taube, Gerson Nakazato, Laila Salmen Espindola, Lorena Carneiro Albernaz, Klinger Antônio da França Rodrigues, Marlus Chorilli, Hugo de Campos Braga, Dayane Batista Tada, Sônia Nair Báo, Luís Alexandre Muehlmann, Mônica Pereira Garcia

**Affiliations:** 1Embrapa Agroenergy, Brazilian Agricultural Research Corporation (EMBRAPA), Brasília 70770-901, DF, Brazil; 2Institute of Chemistry, Federal University of Rio de Janeiro (UFRJ), Rio de Janeiro 21941-853, RJ, Brazil; 3Basic and Applied Bacteriology Laboratory, State University of Londrina (UEL), Londrina 86057-970, PR, Brazil; 4Microscopy and Microanalysis Laboratory, Department of Cell Biology, Institute of Biological Sciences, University of Brasília (UnB), Brasília 70910-900, DF, Brazil; 5School of Pharmaceutical Sciences, São Paulo State University (UNESP), Araraquara 14800-901, SP, Brazil; 6Laboratory of Infectious Diseases, Parnaíba Delta Federal University (UFDPar), Parnaíba 64202-020, PI, Brazil; 7Laboratory for the Isolation and Transformation of Organic Molecules, Institute of Chemistry, University of Brasília (UnB), Brasília 70910-900, DF, Brazil; 8Institute of Biodiversity and Forests, Federal University of Western Pará (UFOPA), Santarem 68005-100, PA, Brazil; 9Pharmacognosy Laboratory, University of Brasília (UnB), Brasília 70910-900, DF, Brazil; 10Institute of Science and Technology, Federal University of São Paulo (UNIFESP), São Jose dos Campos 12231-280, SP, Brazil; 11Faculty of Ceilandia, University of Brasília (UnB), Brasília 72220-900, DF, Brazil; 12Nanobiotechnology Laboratory, Institute of Biological Sciences, University of Brasília (UnB), Brasília 70910-900, DF, Brazil

**Keywords:** Amazon, guarana, AgNPs, biological activities, nanocatalysis, seasonality

## Abstract

**Background:** *Paullinia cupana* Kunth, popularly known as guarana, a native Amazonian shrub cultivated by the Sateré-Mawé ethnic group, has been used in traditional medicine for various purposes, including stimulant and therapeutic actions, due to its chemical composition, which is rich in bioactive compounds. This study explored the reductive potential of guarana with nanobiotechnology and aimed to synthesize silver nanoparticles (AgNPs) using the aqueous extract of leaves collected during the dry and rainy seasons, assessing their biological and catalytic activities. **Methods:** The AgNPs were synthesized in a water bath at 70 °C for three hours and then characterized using techniques such as UV-Vis spectroscopy, DLS, zeta potential, MET, NTA, and EDX and had their effects on various biological systems assessed in vitro, as well as in catalytic tests aimed at indicating the probable influence of the time when the plant material was collected on the properties of the nanostructures. **Results:** The AgNPs had an average diameter between 39.33 and 126.2 nm, spherical morphology, absorption bands between 410 and 450 nm, and high colloidal stability over two years. The biological results showed antibacterial activity against all the species tested, as well as remarkable antioxidant action against DPPH and ABTS free radicals, in the same way as the aqueous leaf extracts of *P. cupana*, in addition to cytotoxic properties against cancerous (A431 and A549) and non-cancerous (HaCaT and HNTMC) cells. The AgNPs were active against promastigote forms of *Leishmania (Leishmania) amazonensis* while not affecting the viability of macrophages, and from the LC_50_ and LC_90_ values, the AgNPs were more effective than the metal salt solution in controlling *Aedes aegypti* larvae and pupae. We also reported that the catalytic degradation of the organic dyes methylene blue (MB) and methyl orange (MO) by AgNPs was over 90% after 40 or 14 min, respectively. **Conclusions:** Thus, our results support the potential of seasonal extracts of guarana leaves to produce AgNPs with diverse application possibilities for the health, industrial, and environmental sectors.

## 1. Introduction

The Amazon rainforest is home to a great deal of plant biodiversity, and guarana (*Paullinia cupana* Kunth), which belongs to the Sapindaceae family, stands out as one of the most promising native compounds in this region of Brazil. The plant’s use by traditional Sateré-Mawé people, mainly in the form of a drink, due to its stimulating and therapeutic effects, has been reported for a long time [1,2]. Because of this, guarana has been listed in the Brazilian Pharmacopoeia since 1977, and in 2005 it was recognized as a safe food by the National Health Surveillance Agency (ANVISA) and the Food and Drug Administration (FDA) [3,4].

The phytochemical composition of *P. cupana* extracts has been reported in some studies, showing differences in the molecular profile depending on the conditions of analysis. When evaluating the different parts of the plant, it is possible to report the occurrence of alkaloids, procyanidins, tannins, and flavonoids in the flower extract, while theobromine and caffeine have been detected in the plant’s nectar [5]. Other studies have described the presence of caffeine and tannins in guarana seeds, as well as methylxanthines in the leaves and pericarp, which stimulates new approaches to genetic improvement and pharmaceutical applications with a focus on different plant organs [6,7]. In addition, the phytochemical composition of guarana seeds from different geographical regions, such as the states of Amazonas, Bahia, and Mato Grosso, showed that dimers and trimers of procyanidins A and B, as well as catechin and epicatechin, were the main chemical markers [8,9]. In addition, higher levels of carbohydrates, caffeine, and tannins have been reported in plant samples from Bahia, while guarana from Amazonas has a high level of fatty acids and cyanolipids [10].

Nanotechnology applies scientific knowledge from various fields, manipulating materials made up of individual atoms, molecules, and molecular clusters to create structures with different or new properties. It is an emerging field, especially in the production and manipulation of nanoscale structures, where generally one of their dimensions must be in the range 1–100 nm [11,12]. Silver is a noble metal with a long history of use in different forms and for various purposes and has long been known for its beneficial properties acting in wound healing and infections, among others [13]. Silver nanoparticles (AgNPs) stand out due to their intrinsic properties such as high stability, strong absorption in the visible ultraviolet region, and broad potential for applications, which are made possible by the significant surface/volume ratio that gives nanoscale particles different attributes to those on a larger scale. In turn, the shape, size, distribution, and surface-related aspects of AgNPs are determined by the concentrations of reducing agents, metal precursors, and stabilizers used during synthesis [14,15].

The search for biocompatible and less toxic methods using natural sources and their derivatives to synthesize AgNPs with improved properties is desirable to maintain better conditions for both the environment and human health [16]. In this scenario, various parts of the same plant can be used to synthesize AgNPs, such as the leaf, bark, flower, seed, fruit, rhizome, and stem, with leaves being the most used materials due to the high volume of extraction, as well as the greater accumulation of biomolecules in their composition [17,18]. These secondary metabolites have the ability to protect plants in the control of various diseases [19], as well as having the ability to chelate with silver ions and subsequently act in the bioreduction and stabilization of AgNPs (Ag^+^ → Ag^0^), binding to their surface and preventing aggregation [20]. While the detailed mechanisms of this process are not yet fully understood, they are driving research into natural products and green nanobiotechnology.

In general, plant metabolism and, consequently, the presence/concentration of plant biomolecules can vary depending on biochemical, physiological, ecological, and evolutionary processes, including changes related to cultivation methods, extraction conditions, planting regions, seasonality, and environmental and genetic factors, among other characteristics such as the age of the plant, altitude, ultraviolet radiation, luminosity, and atmospheric pollution [21]. Thus, the aim of this study was to investigate the potential of *P. cupana* leaves used in the green synthesis of AgNPs by collecting the plant material in seasonal periods of drought and rain in the Amazon region, as well as to assess the possible differences in the characteristics of the nanostructures obtained and their effects on different biological systems in vitro and in the catalytic degradation of organic dyes in order to add value to the sustainable use of natural resources from Brazilian biodiversity.

## 2. Materials and Methods

### 2.1. Chemicals and Reagents

All chemical reagents were used without further purification. Silver nitrate (AgNO_3_), DPPH (2,2-diphenyl-1-picrylhydrazyl), ABTS (2,2′-azino-bis(3-ethylbenzothiazoline-6-sulfonic acid)), fetal bovine serum, penicillin/streptomycin antibiotic solution, methylene blue and methyl orange dyes, and phosphate-buffered saline (PBS) were obtained from Sigma-Aldrich (St. Louis, MO, USA). Methanol (CH_3_OH) and dimethyl sulphoxide (DMSO) were purchased from Dinamica (Indaiatuba, São Paulo, Brazil). Potassium persulphate (K_2_S_2_O_8_) was purchased from LabSynth (Diadema, São Paulo, Brazil). Sodium borohydride (NaBH_4_) was purchased from Alphatec (São Paulo, Brazil).

The bacteria *Acinetobacter baumannii*, *Escherichia coli*, *Klebsiella pneumoniae*, *Pseudomonas aeruginosa*, *Salmonella enterica*, *Staphylococcus aureus*, and *Staphylococcus epidermidis* were obtained from the American Type Culture Collection (ATCC), while the bacterium *Bacillus cereus* was isolated from the environment and kept in the laboratory. The Muller–Hinton agar culture medium was obtained from Difco (Difco, Franklin Lakes, NJ, USA).

The cell lines were acquired from the Rio de Janeiro Cell Bank (BCRJ) (Rio de Janeiro, Brazil). DMEM (Dulbecco’s Modified Eagle Medium) culture medium and MTT (3-(4,5-dimethyl-2-thiazolyl)-2,5-diphenyl-2H-tetrazolium bromide) salt were purchased from Thermo Fisher Scientific (Waltham, MA, USA). Only ultrapure water was used to prepare the plant extracts, the metal salt solution, and for other purposes during the study.

### 2.2. Preparation of Aqueous Extract of Paullinia cupana Leaves and Synthesis of AgNPs

The collection of plant material and the methodology used to prepare the aqueous extract of *P. cupana* leaves were carried out as described in our previous studies [22,23], with the aqueous extracts being called Ext-LD (aqueous extract of leaves collected during the dry season) and Ext-LR (aqueous extract of leaves collected during the rainy season). The biogenic synthesis of AgNPs was carried out using an aqueous solution of AgNO_3_ at 2 mM (340 µg/mL) and leaf extract at a concentration of 2 mg/mL in a final volume of 50 mL. In addition, experimental controls were also produced consisting of the following: (i) AgNO_3_ control—2 mM metal salt and ultrapure water and (ii) extract control—2 mg/mL aqueous extract and ultrapure water, both in a final volume of 50 mL. The flasks containing each solution were protected from light with aluminum foil and incubated in a water bath (555, Fisatom, São Paulo, Brazil) at 70 °C for 180 min. The obtained suspensions were named AgNPs-LD (nanoparticles synthesized using extract from leaves collected during the dry season) and AgNPs-LR (nanoparticles synthesized using extract from leaves collected during the rainy season).

### 2.3. Characterization of AgNPs

#### 2.3.1. Ultraviolet–Visible Spectroscopy (UV-Vis)

The kinetics of AgNPs formation were monitored in a UV-Vis spectrophotometer (UV1800PC, Phenix, Blomberg, Germany) every 30 min during incubation in a water bath. Then, 2 mL aliquots of each sample, without prior dilution, were placed in a quartz cuvette and inserted into the equipment, and their absorbance was measured at a wavelength of 450 nm. At the end of the reactions, with the same volume of colloidal suspension, spectrophotometric analyses were carried out in the range between 350 and 550 nm to obtain the maximum absorption bands of each sample.

#### 2.3.2. Dynamic Light Scattering (DLS) and Surface Zeta Potential

The colloidal characteristics of average hydrodynamic diameter (HD), polydispersity index (PdI), and surface zeta potential (ZP) of the AgNPs were measured using a ZetaSizer Nano ZS device (Malvern Instruments, Malvern, UK). For these analyses, the samples were initially diluted in a 1:10 (*v*/*v*) ratio in ultrapure water to a final volume of 1 mL, and then this solution was placed in a polystyrene cuvette to be inserted into the equipment, which was configured to take the readings at an angle of 90°, a temperature of 25 °C, with 120 s of stabilization, and readings in triplicate with 10 runs each.

To analyze the stability of the AgNPs, aliquots were separated into two storage conditions: at room temperature (25 °C) and under refrigeration (4 °C). Analyses were carried out over 730 days (2 years) from the initial synthesis of the nanostructures, and the data were represented as the mean ± standard deviation, processed using the *ZetaSizer* 7.13 software developed by the same equipment manufacturer.

#### 2.3.3. Nanoparticle Tracking Analysis (NTA)

To determine the concentration, size and *Span* index of the AgNPs, a methodology adapted from a previous study was used [24]. Nano-Sight NS300 equipment (Malvern Instruments, Worcestershire, UK) equipped with a 532 nm diode laser (green) was used where the samples, diluted in a ratio of 1:100 (*v*/*v*) in ultrapure water to obtain a concentration of between 1 × 10^2^ and 10^8^ particles/mL, were injected with sterile syringes, and the captured videos were analyzed using NTA 3.2 software (Malvern Instruments, Malvern, UK).

The average size values obtained by the analysis software were based on the values calculated with the sizes of all the particles captured by the equipment and used to determine *D*10, *D*50, and *D*90 as average size distributions. The *Span* index, which denotes the dispersity of the populations of particles in suspension, was calculated using Equation (1). In turn, the results were presented as the mean ± standard deviation and the mode ± standard deviation from analyses carried out in triplicate for each type of AgNPs.(1)$$Span=(\frac{D90-D10}{D50})$$

#### 2.3.4. Transmission Electron Microscopy (TEM)

The morphological characterization of the AgNPs was carried out using a JEM-1011 transmission electron microscope (JEOL, Tokyo, Japan) operating at 80 kV. Briefly, 5 µL of each sample, without dilution, were placed on a 400-mesh copper screen covered with Formvar^®^ film, and after drying for 24 h at room temperature, the nanostructures were visualized in the equipment.

Photomicrographs were taken at random and from the particle count of each group of AgNPs (between 340 and 450 particles) using *ImageJ* software version 1.8.0 (National Institute of Health, Bethesda, MD, USA). Size distribution histograms were constructed showing the mean ± standard deviation of the mean dry particle diameter.

#### 2.3.5. Energy Dispersive X-Ray Spectroscopy (EDX)

The elemental composition of the atoms present in the colloidal suspensions was determined by EDX analyses carried out by viewing the samples in a scanning electron microscope (FEI, INSPECT S50 with Everhart-Thornley detector, Hillsboro, OR, USA) operating at 10 kV. Each AgNPs sample was placed separately on aluminum stubs without metallization and left in the dark for three days in closed containers until completely dry for subsequent analysis in the equipment.

### 2.4. Biological Studies of AgNPs

#### 2.4.1. Antibacterial Activity

The broth microdilution method according to the Clinical Laboratory Standard Institute [25] was used for this test. The following bacteria were used Gram-negative: *Acinetobacter baumannii* (ATCC 19606), *Escherichia coli* (ATCC 25922), *Klebsiella pneumoniae* (ATCC 700603), *Pseudomonas aeruginosa* (ATCC 9027), and *Salmonella enterica* (ATCC 13076). The following bacteria were used Gram-positive: *Bacillus cereus* (environmentally isolated), *Staphylococcus aureus* (ATCC 25923), and *Staphylococcus epidermidis* (ATCC 12228).

Initially, to assess the minimum inhibitory concentration (MIC), the bacteria were seeded on Muller–Hinton (MH) agar and incubated for 24 h at 37 °C. After this period, around five bacterial colonies were collected and placed in saline solution (0.85% NaCl) with the *McFarland* scale adjusted to 0.5 or 1.5 × 10^8^ CFU/mL. Then, 50 µL of MH was added to the wells of a 96-well microplate, and 50 µL of each sample was added afterwards, totaling 100 µL. Next, 50 µL of this solution was collected and placed in the next well and so on, carrying out a serial dilution with seven dilution points for the AgNPs and AgNO_3_ (42.5 to 0.33 µg/mL) and for the aqueous extracts (25,000 to 50 µg/mL). At the end, to complete the final volume of 100 µL per well, the saline solution with the bacteria was diluted to 1.5 × 10^6^ CFU/mL, and 50 µL were added to each well, to which the sample to be tested had previously been added.

The plates were incubated for between 18 and 24 h at 37 °C and then read visually, assessing the turbidity of the wells, where the absence of turbidity indicated inhibition of bacterial growth. All the tests were carried out in triplicate per concentration, with a positive control (only saline solution of bacteria) and a negative control (only MH medium). The minimum bactericidal concentration (MBC) was considered when bacterial death was ≥99.9% after 24 h of incubation.

#### 2.4.2. Antioxidant Activity Against DPPH and ABTS Free Radicals

To determine antioxidant activity, 150 µL of a methanolic solution of the DPPH radical (0.1 mM) was added to a 96-well microplate, followed by 150 µL of each sample at different concentrations, with AgNPs (12.5, 25, 50, 75, 100, and 150 µg/mL), aqueous extracts (125, 250, 500, 750, 1000, and 1500 µg/mL), and ascorbic acid (positive control) at the same concentrations as the nanostructures, and then the microplates were covered with aluminum foil and incubated in the dark at room temperature for 30 min [26]. After this period, the absorbance of the wells was read at 517 nm using a spectrophotometer combined with a microplate reader (M3, Molecular Devices, San Jose, CA, USA). For this experiment, the negative control consisted of 150 µL of methanolic DPPH solution with 150 µL of methanol, and 300 µL of methanol was used as an experimental blank.

The antioxidant activity of the samples was also tested against the ABTS^+^ radical cation, formed from the reaction between aqueous solutions of ABTS (7 mM) and potassium persulphate (140 mM), according to previous studies with modifications [27,28]. In a 96-well microplate, 250 µL of the aqueous ABTS^+^ solution was deposited in the wells, followed by 50 µL of AgNPs (12.5, 25, 50, 75, 100, and 150 µg/mL), aqueous extracts (125, 250, 500, 750, 1000, and 1500 µg/mL), and ascorbic acid (positive control) at the same concentrations as the nanostructures. The microplates were then covered with aluminum foil and incubated for 7 min in the dark at room temperature, and then the absorbance of the wells was read at 734 nm on a spectrophotometer combined with a microplate reader (M3, Molecular Devices). The negative control was 250 µL of aqueous ABTS^+^ solution with 50 µL of potassium persulphate, and the experimental blank was 300 µL of potassium persulphate.

All experiments were carried out in triplicate per concentration for each sample and in three independent experiments. The results are presented as the mean ± standard deviation of the mean, with the antioxidant activities expressed as a percentage of free radical *inhibition* (%) based on the absorbance values corrected for their respective blanks. The antioxidant activities were calculated using Equation (2).(2)$$Inhibition\;(\%)=\frac{\left(Absorbance\;of\;the\;control-Absorbance\;of\;the\;sample\right)}{Absorbance\;of\;the\;control}\times 100$$

#### 2.4.3. Anticancer Activity

The anticancer activity of AgNPs, AgNO_3_, and aqueous extracts was tested on A431 (non-melanoma skin carcinoma), HaCaT (human keratinocyte), A549 (lung adenocarcinoma), and HNTMC (human dental pulp fibroblast) cells using MTT colorimetric assay. The cell lines were grown in DMEM medium supplemented with 10% (*v*/*v*) fetal bovine serum (FBS) and 1% (*v*/*v*) antibiotic solution (100 U/mL penicillin and 100 µg/mL streptomycin) and kept in cell culture flasks in an incubator (Thermo Scientific, Waltham, MA, USA) at 37 °C with 80% humidity and 5% CO_2_.

A431, HaCaT, and A549 cells were plated at a density of 1 × 10^4^ cells/well, while HNTMC cells were plated at 5 × 10^3^ cells/well and incubated under the same conditions described above for 24 h at 200 µL per well in 96-well microplates. After this period, the supernatant was removed, and the cells were treated with 200 µL of new culture medium containing different concentrations of the samples, namely AgNPs and AgNO_3_ (2, 4, 6, 8, 10, and 20 µg/mL), as well as the aqueous extracts (200, 400, 600, 800, 1000, and 2000 µg/mL). The negative control was prepared only with supplemented culture medium without any treatment.

The MTT test was carried out after 24 h of incubating the cells with the experimental samples. Initially, the contents of the wells were discarded, and they were washed with PBS and then received 150 µL of 10% MTT solution (0.5 mg/mL) diluted in culture medium. The plates were incubated for 2 h under the same conditions as above, this time covered in aluminum foil. After this period, the supernatant from the wells was removed and replaced with 150 µL of DMSO to dissolve the formazan crystals that had formed. The contents of the wells were homogenized and, after five minutes, the absorbance was measured at 595 nm on a spectrophotometer combined with a microplate reader (Multiskan FC, Thermo Scientific, Waltham, MA, USA).

Three independent experiments were carried out for each cell line, with repetitions in quadruplicate per concentration for each sample. The results obtained were expressed as a percentage of cell viability (%) based on the absorbance values corrected for the control (wells with supplemented DMEM medium, without any treatment, with *cell viability* set at 100%) and presented as the mean ± standard deviation of the mean, according to Equation (3), as follows:(3)$$Cell\;viability\;(\%)=(\frac{Absorbance\;of\;the\;sample}{Absorbance\;of\;the\;control})\times 100$$

#### 2.4.4. Antileishmanial Activity, Cytotoxicity, and Selectivity Index (SI)

Parasites of the species *Leishmania (Leishmania) amazonensis* (IFLA/BR/67/PH8) in promastigote form were used to determine antileishmanial activity. The cells were grown in supplemented Schneider’s medium (10% fetal bovine serum and 1% solution containing penicillin—100 U/mL—and streptomycin—100 ug/mL) at pH 7 and maintained at 26 °C in a biochemical oxygen demand (BOD) incubator (Eletrolab EL202, São Paulo, Brazil) with weekly relays [29].

The promastigote culture in logarithmic growth phase was grown in a 96-well microplate with 1 × 10^6^ parasites/well in 100 µL of Schneider’s medium containing varying concentrations of the samples (AgNPs, AgNO_3_, or aqueous extracts) obtained by serial dilution from 100 to 1.56 µg/mL in four independent experiments in triplicate. The positive control was the drug miltefosine at the same concentrations as the AgNPs, and the negative control was the supplemented culture medium containing 0.2% DMSO (considered to be 100% viability of the leishmania). The plates were incubated in a BOD for 72 h at 26 °C. At the end of this time, 10 µL of MTT (5 mg/mL) was placed in each well, and the plates were then incubated for 4 h. After this period, the supernatant was removed from the wells, which received 50 µL of 10% sodium dodecyl sulphate (SDS) solution (*w*/*v* in distilled water) to solubilize the formazan crystals. The absorbance of the wells was measured in a microplate spectrophotometer at 540 nm (Biosystems ELx800, Curitiba, PR, Brazil).

For the cell viability tests to determine the selectivity index, the RAW 264.7 murine macrophage strain was used, which was maintained in cell culture flasks (Corning Glass Workers, New York, NY, USA) in supplemented DMEM medium (10% SFB, 1% antibiotic—100 U/mL penicillin and 100 μg/mL streptomycin), pH 7, at 37 °C, with 5% CO_2_ and 80% humidity. Recipes were carried out after the cells reached confluence, characterized by the formation of a cell monolayer around 48 to 72 h after the initial incubation [30].

Cytotoxicity was assessed in 96-well microplates in which 100 μL of DMEM medium supplemented with 1 × 10^5^ macrophages were placed per well. After 4 h of incubation, the wells were washed three times with sterile PBS, and then 100 μL of DMEM medium supplemented with different serial concentrations of AgNPs, AgNO_3_, aqueous extracts (200 to 3.12 µg/mL), and the drug miltefosine (400 to 6.25 µg/mL) were added, followed by incubation for 72 h. At the end of the period, 10 μL of MTT (5 mg/mL) was applied to the wells, and the plates were incubated again for a further 4 h. The supernatant was then removed, and 100 μL of DMSO was added to each well. After 30 min of stirring, the absorbance was read at 540 nm on a plate reader (Biosystems ELx800, Curitiba, PR, Brazil). Only DMEM-supplemented medium containing 0.5% DMSO was used as a negative control (considered to possess 100% macrophage viability).

The selectivity index (SI) of each test sample was calculated by dividing the 50% cytotoxic concentration for macrophages (CC_50_) by the maximum concentration capable of inhibiting the growth of promastigote forms of leishmania by 50% (CI_50_).

#### 2.4.5. Insecticidal Activity Against *Aedes aegypti* Larvae and Pupae

The colonies of *Aedes aegypti* (Rockefeller strain) used for the tests were kept in the laboratory, without exposure to any insecticide, under the following controlled conditions: temperature 28 °C (±2 °C), 70% (±10%) relative humidity, and 12/12 h photoperiod (light/dark). The eggs hatched in plastic trays containing tap water, and the larvae were fed fish food (0.5 g) every day. The newly formed pupae were separated into males and females (in a 1:3 ratio, respectively) and transferred to other trays. The adult insects were fed a 10% sugar solution, and the females were given equine blood (provided by the Veterinary Hospital of the University of Brasilia) three times a week for egg production and maturation.

Larvicide tests were carried out in accordance with the recommendations of the World Health Organization (WHO) [31] with modifications to determine the lethal concentration (LC_50_ and LC_90_). Briefly, 25 third instar larvae (L3) aged between 72 and 96 h were transferred to 200 mL transparent plastic cups containing different concentrations of AgNPs and AgNO_3_ (0.1, 0. 5, 2, 5, 10, 15, and 20 µg/mL), as well as the aqueous extract (200, 500, 1000, 1500, and 2000 µg/mL). The final volume was 120 mL/cup topped up with tap water, and the negative control was only tap water in the same volume as the cups with the test samples. Four independent experiments were carried out on different batches of larvae in quadruplicate for each sample tested, and larval mortality was assessed after 24, 48, and 72 h of exposure, with larvae that did not move and/or did not react to stimuli after gentle shaking of the beaker being considered non-living.

The pupicidal tests to determine the LC_50_ and LC_90_ values were carried out on 10 newly emerged pupae, no more than one day old, which were transferred to 50 mL plastic cups and then covered with a fine mesh. Different concentrations of AgNPs (0.156, 0.625, 2.5, 5, 10, and 20 µg/mL) and AgNO_3_ (0.468, 1.87, 3.75, 7.5, 15, and 30 µg/mL) were added to each cup, and the final volume was topped up to 20 mL/cup with tap water. Mortality was assessed after 24 and 48 h of exposure to the samples (the time limit for the pupae in the control not to turn into mosquitoes), and pupae that did not move and/or did not react to stimuli after lightly shaking the beaker were considered not to be alive. Only tap water in the same volume as the beakers containing the test samples was used as a negative control. Four independent experiments were carried out in quadruplicate per test sample, each using different batches of pupae.

### 2.5. Catalytic Activity

The catalytic activity of AgNPs was evaluated in the presence of sodium borohydride (NaBH_4_) as a substrate for the reduction of methylene blue (MB) and methyl orange (MO) dyes at room temperature. For the test with MB (λ = 664 nm), freshly prepared aqueous solutions of the dye were placed in a quartz cuvette (2.5 mL; 0.08 mM), plus NaBH_4_ (500 µL; 60 mM), and then each AgNPs suspension (500 µL; 100 µg/mL) was added individually in separate experiments, with the degradation reactions monitored using absorbance readings obtained by a UV-Vis spectrophotometer (UV-2600, Shimadzu, Kyoto, Japan) in the 200–800 nm range at five-minute intervals [32]. For the MO tests (λ = 464 nm), a reaction mixture was prepared in a quartz cuvette with aqueous solutions of the organic dye (2.5 mL; 0.08 mM) and NaBH_4_ (500 µL; 60 mM), both freshly prepared, in addition to the individually arranged AgNPs suspensions (20 µL; 20 µg/mL), and immediately afterwards, the absorbance recordings were started on a UV-Vis spectrophotometer (UV-2600, Shimadzu, Kyoto, Japan) in the 200–700 nm range with readings at two-minute intervals [33].

During the contact time between the nanocatalysts with the dye and substrate, there is a process of decolorization of the reaction mixtures, and the end of the reactions was defined by the transparent appearance of the solutions and the decrease in the intensity of the absorption bands characteristic of each dye. Control experiments were also carried out as mentioned above, but with the volume of nanocatalysts replaced by ultrapure water. Thus, with the absorbance data obtained at each reading time, it was possible to calculate the percentage *degradation* of each pollutant according to Equation (4), and the chemical kinetics of the reactions were obtained using three kinetic models: zero order, 1st order, and 2nd order, according to Equations (5)–(7), respectively [34].(4)$$\%\;degradation=\frac{\left(A₀-Aₜ\right)}{A₀}\times 100$$(5)A₀−Aₜ=kt(6)$$\ln\left(\frac{Aₜ}{A₀}\right)=-kt$$(7)$$\frac{1}{A_{t}}-\frac{1}{A_{0}}=kt$$
where *k* is the rate constant of the reaction, *t* is the final reaction time, *A*_0_ is the initial absorbance of the pollutants at time 0, and *A_t_* is the absorbance of the pollutants at time *t*. *A*_0_ and *A_t_* are the absorbances measured at 664 and 464 nm for MB and MO, respectively, at time zero and at the aforementioned UV-Vis reading intervals during each reaction. Based on the curves constructed using linear regression, the *k* values were determined, as well as the correlation coefficients (R^2^), and the most appropriate kinetic model was chosen for each reaction involving the AgNPs, the substrate, and the pollutants tested.

### 2.6. Statistical Analysis

Statistical analyses of the DLS data (DH, PdI, and ZP) were carried out using a one-way ANOVA test followed by the Tukey test, with a significance level of 95% (*p* < 0.05). The CI_50_ values in the antioxidant, anticancer, and antileishmanial activity tests, as well as the CC_50_ data in the antileishmanial test and the LC_50_ and LC_90_ values in the insecticidal activity tests, were determined from a sigmoidal dose-dependent response by non-linear regression with four parameters, based on the normalized logarithmic analysis graphs (log (inhibitor) vs. variable slope) from the normalization of the values, transformation into log values, and then the calculation to define the 50% concentrations and R^2^ values. In these tests, the possible statistical differences between the experimental groups were based on one-way ANOVA (antileishmanial, larvicidal, and pupicidal) or two-way ANOVA (antioxidant and anticancer) tests, followed by Tukey or Bonferroni tests (specifically for the antileishmanial test) with a significance level of 95% (*p* < 0.05).

For all these analyses, the *GraphPrism* 8 statistical program (GraphPad Software, San Diego, CA, USA) was used for plotting and constructing the graphs/histograms, except for the construction of the histogram obtained by TEM and the graphs of the catalytic experiment in which the *OriginPro* 8.5 software (OriginLab Corporation, Northampton, MA, USA) was used.

## 3. Results and Discussion

### 3.1. Visual Appearance and UV-Vis Spectrophotometry

Initially, the biogenic synthesis of the AgNPs was evaluated by means of the color change in the reaction medium, as illustrated in Figure 1, and among the effects observed, it is possible to notice the more intense shade of yellow in the tubes containing the AgNPs.

This phenomenon, characterized by the change in color of colloidal suspensions, is known as Tyndall scattering and occurs from visual optical changes that are observed when light is deflected due to the reflection caused by the nanostructures suspended in an aqueous medium [35]. In addition, variations in color depend on the season in which the plant material was collected but also on the conditions adopted during synthesis, including the concentration of the metallic precursor agent, the concentration of the biological extract, time, and temperature [36].

Kinetic monitoring of the formation of AgNPs indicated that the aqueous extracts of *P. cupana* leaves have the potential to reduce metal ions and form nanostructures, regardless of when the plant material was collected. As shown in Figure 2, AgNPs-LR showed greater absorption intensity (1.572 a.u.) after 180 min of incubation compared to AgNPs-LD (0.648 a.u.), which tended to stabilize at the end of the synthesis time, while the experimental controls showed no change.

UV-Vis analysis made it possible to identify the spectral band characteristic of each type of nanostructure, and it is also an analysis that makes it possible to confirm the presence of AgNPs since, when they receive radiation, the free electrons oscillate and become excited, causing a phenomenon known as surface plasmon resonance (SPR) [37]. In Figure 3, the AgNPs-LD showed maximum absorption at a wavelength of 410 nm (0.787 a.u.), while a shift to a longer wavelength was observed for the AgNPs-LR with a band at 450 nm (0.700 a.u.), and in the spectra of the experimental controls, no considerable change in absorbance was obtained.

In general, the absorbance spectra varied in the shape and symmetry of the curves, and the shifts in the wavelengths of maximum absorption, as well as the absence of well-defined spectral bands (broadening of the curves) represent, among other things, variations in the size, polydispersity, morphology, and, consequently, colloidal stability of the nanostructures [38]. In addition, the synthesis of AgNPs mediated by biological organisms can be affected by geographical and/or seasonal variations, resulting in different properties, as observed in studies in which AgNPs showed an absorbance band between 426 and 434 nm after using extracts of the plants *Sisymbrium irio* and *Cleome viscosa* collected in different seasons of the year [39,40].

### 3.2. Evaluation of Colloidal Stability by DLS and Surface Zeta Potential Analysis

The probable changes in the stability of AgNPs, such as their dimensional and electrical characteristics, are not desirable in biogenic synthesis, especially after long periods under different storage conditions. Table 1 and Table 2 show the HD, PdI, and ZP values of AgNPs-LD and AgNPs-LR, respectively, monitored for 730 days (2 years) under two storage conditions.

Regarding monitoring colloidal stability, the newly synthesized AgNPs-LD had an initial hydrodynamic diameter of 81.90 nm and suffered significant decreases (*p* < 0.05) only in the first two weeks after storage at room temperature. On the other hand, the AgNPs-LR initially measured 91.74 nm, and significant increases (*p* < 0.05) in the samples kept in the fridge were only seen after six months and two years. On the other hand, PdI did not show significant changes throughout the monitoring of the colloidal stability of AgNPs-LD and AgNPs-LR in relation to the measurement at the initial time, which was 0.448 and 0.295, respectively, indicating moderately polydisperse particle populations [41]. The ZP of the samples showed low and/or moderate stability, as recommended [42], showing significant momentary decreases (*p* < 0.05) in the surface charge of the AgNPs-LD, which was −38.1 mV, and for the AgNPs-LR in the first week and after one year in both storage conditions, in relation to the initial electrical charge of −34.9 mV.

Our results corroborate previous reports on AgNPs synthesized using extracts from *Ceropegia debilis* and *Cymbopogon citratus* leaves collected in summer. These studies reported an average nanoparticle diameter ranging from 88.2 to 100.6 nm, with a PdI between 0.193 and 0.3 [43,44]. On the other hand, the diameter of AgNPs produced from *Olea europaea* leaf extract and propolis collected in winter ranged from 12 to 68 nm, with a zeta potential between −15 and −52 mV [45,46].

It is important to note that the differences observed in the stability tests of the AgNPs synthesized with the extract of *P. cupana* leaves may be related to the following: (i) the time of collection/location of the plant material, (ii) the part of the plant used, (iii) the storage environment of the colloidal suspensions, and (iv) the reducing capacity of the metabolites in the plant extracts [8,47]. Our previous study [22] characterized the phytochemical profile of the aqueous extracts of *P. cupana* described here to assess their composition, and the results revealed a high content of phenolic compounds, as well as a high antioxidant potential against free radicals, in addition to the presence of a variety of secondary metabolites that were extracted using the decoction/boiling method, including alkaloids, flavonoids, terpenoids, carboxylic acids, and procyanidins. These bioactive compounds, as reported in previous research, belong to the well-known reducing agent classes of silver ions responsible for the successful synthesis of AgNPs [48,49,50,51,52].

### 3.3. Nanoparticle Tracking Analysis (NTA)

The size of the AgNPs was also analyzed by NTA, and values of 68.5 nm and 89.3 nm were observed for AgNPs-LD and AgNPs-LR, respectively, strongly corroborating the data obtained by DLS. The particle size found most frequently was 62.3 nm and 73.1 nm for the dry and rainy season nanostructures, respectively. In turn, the synthesis process resulted in a suspension of AgNPs-LD containing 5.33 × 10^7^ particles/mL and 1.5 × 10^8^ particles/mL for AgNPs-LR (Table 3).

Regarding the populations of particles formed, unimodal distribution is shown for the AgNPs-LD (Appendix A), while there is a multimodal distribution with populations of varying sizes in the AgNPs-LR sample (Appendix A). This behavior is in line with the results of the *Span* index, which were 0.443 and 0.523, respectively, since the more polydisperse the suspension, the higher this value.

NTA analysis is an important technique often used to estimate the size, distribution, and concentration of nanostructures in liquid media by tracking individual nanoparticles. In a previous study, a natural polysaccharide obtained from *Anacardium occidentale*, a plant found in the northeastern region of Brazil, was used to prepare AgNPs with a hydrodynamic diameter of 51.9 nm and 3.22 × 10^4^ particles/mL [53], and more recently the leaf extract of *Eucalyptus camaldulensis* was used to synthesize AgNPs with an average size of 99 nm and 24.88 × 10^8^ particles/mL [54].

### 3.4. Morphological Analysis by MET and Compositional Analysis by EDX

Figure 4A shows the TEM micrographs of the AgNPs-LD, revealing a particle size distribution between 22.5 and 92.5 nm, with a dry diameter of 39.33 nm, while in Figure 4B the size range of the AgNPs-LR was 17.5–97.5 nm, with a dry diameter of 42.43 nm. The morphology of the samples varied, with spherical, quasi-spherical, rod, triangular, prismatic, and hexagonal shapes, among others.

Overall, the images showed morphologies to be clearly distinguished and confirm the small size of the AgNPs, but it is possible to observe a distribution in the size of the nanostructure populations that may be related to the broad bands shown in the UV-Vis spectra and the moderate PdI/*Span* index values that reflect the polydispersity of the samples [55]. In addition, in common with the images, the stabilization layer surrounding the surface of the AgNPs can be seen, which may refer to the biomolecules in the plant extract [56].

It is important to emphasize that even with the change in seasonal conditions and differences in phytochemical composition between the extracts, no significant changes were observed in the diameters of the AgNPs. Some research has shown that AgNPs synthesized with extracts from medicinal plants collected in summer had a diameter of between 11.36 and 75.7 nm [57,58], while AgNPs produced from citrus and aquatic plant species collected in winter ranged in size from 12 to 58.25 nm [59,60].

The information provided by EDX analysis reveals, based on the normalized weight percentage, the content of the atoms that represent the total composition of the samples by means of the intensity of the peaks obtained and allows the quantification of the chemical elements [61]. According to Figure 5A, the proportion of silver in the AgNPs-LD spectrum is 18.03%, and in Figure 5B, the silver content in AgNPs-LR is 26.5%, with both showing dispersion peaks at the start of the spectra and around 3 keV, typical of metallic silver [62].

The intense peak observed at around 1.5 keV indicates the presence of aluminum from the stub where the samples were previously deposited for reading on the equipment, and the other signals, referring to magnesium (Mg), silicon (Si), carbon (C), and oxygen (O) atoms, may come from the layer of biomolecules covering the surface of the nanostructures, as well as appearing as traces of the mineral content of the soil where the plant is grown and which may be absorbed by the plant tissues [63,64].

### 3.5. Antibacterial Activity

In this study, the MIC results showed that AgNPs were effective against all bacterial species, with values between 10.60 and 42.50 µg/mL (Table 4). The MBC values were between 5.30 and 42.50, equal to or at least twice as high as the MIC concentrations obtained. In relation to the results for AgNO_3_, it was seen that, in general, the inhibitory effect on bacterial growth was greater than that observed after exposure to AgNPs, and in fact, due to their small size (around 0.26 nm), the bioavailability and internalization of Ag^+^ ions by bacterial cells can be increased [65].

Thus, it is possible to infer that there is no direct correlation between the two classes of AgNPs tested that could indicate which sample is more efficient against bacteria, but it is suggested that these nanostructures may be suitable for use as bacterial decontamination agents. Furthermore, it is worth pointing out that based on the MIC and MBC values, susceptibility was similar for both classes of bacteria tested (Gram-positive and Gram-negative).

The antimicrobial activity of silver has been known since antiquity and is interesting due to its high toxicity to microorganisms and low toxicity to human cells [66]. It is worth noting that the mechanism of action of Ag^+^ ions has been elucidated previously, but there is no consensus on the mechanisms related to the antibacterial effects of AgNPs, which, in general, are related to the small diameter combined with the high surface area of the nanostructures, which facilitates permeation into bacterial membranes [67,68].

Another important factor is the electrostatic attraction that can occur due to interactions between positively charged metal ions and the bacterial cell membrane, which is formed by molecules containing negatively charged functional groups (thiol, sulfhydryl, imidazole, amino, phosphate, teichoic acids, and lipopolysaccharides), which can lead to alterations in protein structure, changes in membrane permeability, the generation of reactive oxygen species (ROS), and the blocking of ATP synthesis, preventing cell division and causing the death of bacteria [44,69]. Furthermore, even if AgNPs have a negative charge, as in this study, bactericidal activity can occur since these nanostructures interact with sulfur-containing amino acids (cysteine, homocysteine, methionine, and taurine) causing damage to the bacterial membrane [70,71].

The aqueous extracts of *P. cupana* leaves showed no antibacterial action against any of the species tested similarly to what was reported in a recent study in which the extract of guarana seed hulls showed no activity against various bacteria [72]. However, other studies have used extracts of *P. cupana* seeds and obtained antibacterial efficacy resulting in MIC and MBC of 250 µg/mL against Gram-positive and Gram-negative bacteria [73], as well as MICs in the range of 16–128 µg/mL and an inhibitory effect of between 2.1 and 100% against various bacterial species [74,75].

### 3.6. Antioxidant Activity

As shown in Figure 6, the ability of the nanostructures to eliminate DPPH radicals increased in a dose-dependent manner, and at the highest concentration tested (150 µg/mL), the inhibition caused by AgNPs-LD and AgNPs-LR was 50.89% and 66.86%, respectively, with an IC_50_ of 43.16 and 36.93 µg/mL. In turn, the positive control ascorbic acid (AA) showed an elimination capacity of over 95%, with an IC_50_ of 22.43 µg/mL. After incubation with Ext-LD, the antioxidant activity ranged from 22.46 to 90.71% in the concentration range (125–1500 µg/mL), with an IC_50_ of 476.4 µg/mL, and for Ext-LR, the free radical scavenging was 93% from the concentration of 500 µg/mL, with an IC_50_ of 229.7 µg/mL.

Figure 7 shows the ABTS radical inhibition results after exposure to the experimental groups, showing a dose-dependent behavior in which free radical elimination was similar when analyzed within the concentration range for AgNPs-LD (58.53–66.72%) and AgNPs-LR (55.48–73.62%), with an IC_50_ of 54.2 and 61.19 µg/mL, respectively. Inhibition of AA ranged from 69.95 to 100%, resulting in an IC_50_ of 24.98 µg/mL, and in relation to radical scavenging by Ext-LD, at a concentration of 1500 µg/mL, inhibition was 74.62%, with an IC_50_ of 414.5 µg/mL, while Ext-LR promoted elimination by 92.77% at the highest concentration, with an IC_50_ of 445.3 µg/mL.

Our results show greater efficacy compared to previous research using *Eupatorium adenophorum* and *Oxalis corniculata* leaf extracts for the synthesis of AgNPs, in which 48.96 and 94.3 µg/mL were required to eliminate 50% of DPPH and ABTS radicals, respectively [76,77]. In turn, the methanolic extract of guarana seeds showed a strong DPPH radical scavenging capacity ranging from 83.4% to 90.9% [78], while another study estimated the antioxidant capacity of extracts from different parts of guarana and obtained 9534 µM eq. Trolox/g for the seed and 68 µM eq. Trolox/g for the bark against ABTS radicals [79].

Free radicals are chemical species with unpaired electrons, making them highly unstable and thus reactive in chemical reactions with other molecules. They can be derived from oxygen, nitrogen, and sulfur, for instance, creating distinctive reactive species which, in excess, can trigger various diseases since they mainly target proteins, DNA, and RNA [80,81]. In this sense, to nullify the harmful effects caused by oxidative stress, cells have developed an antioxidant defense machinery that involves enzymes (peroxidase, superoxide dismutase) and reducing agents, such as phenolic compounds [82]. The mechanism of action in which AgNPs are involved is mainly based on the transfer of free electrons and by donating hydrogen atoms to the free radical molecules, which act as acceptors and become reduced, with more stable structural characteristics and a change in the color of the final solution [83,84].

### 3.7. Cytotoxic Activity

The cytotoxic effects caused after exposures of the different cell lines to AgNPs and AgNO_3_ show a dose-dependent behavior within the range of concentrations tested, as well as non-specific activities, since the samples were effective on both cell types (cancerous and non-cancerous). In A431 cells, at the highest concentration (20 µg/mL), less than 20% of cells remained alive after incubation with the samples, and the IC_50_ values were 5.122 µg/mL (AgNO_3_), 5.510 µg/mL (AgNPs-LR), and 5.851 µg/mL (AgNPs-LD) (Figure 8A; Appendix A). On the other hand, against the HaCaT strain, there was a considerable reduction in viability from the intermediate concentration of 6 µg/mL, resulting in less than 8% of viable cells and an IC_50_ between 3.497 and 3.940 µg/mL (Figure 8B; Appendix A).

The viability of A549 cells after incubation with the samples was reduced from 7% to 5% at the highest test concentration, with AgNPs-LD, AgNPs-LR, and AgNO_3_ having an IC_50_ of 5.436, 5.609, and 4.750 µg/mL, respectively (Figure 8C; Appendix A). For the HNTMC lineage, high susceptibility was observed from the concentration of 6 µg/mL onwards, as less than 15% of the cells remained alive, resulting in an IC_50_ of 3.490–4.231 µg/mL (Figure 8D; Appendix A).

Among the mechanisms proposed for the cytotoxic activity of AgNPs are oxidative stress, which may be responsible for the increase in pro-apoptotic genes [85,86], alterations in mitochondrial membrane potential, and the interruption of ATP synthesis through interaction with functional groups on the cell surface, generating free radicals [87,88]. In addition, after incubation with the nanostructures, the cell cycle may be interrupted in different phases. This leads to a reduction in angiogenesis and cell proliferation, preventing metastases [89,90]. Other intrinsic aspects of AgNPs allow for cytotoxic action, such as particle size [91], electrical charge [92], morphology [93], and surface coating [94].

As shown in Figure 8 and Appendix A, the aqueous extracts of *P. cupana* leaves did not significantly affect cell viability. After inoculation with the A431 and A549 cell lines, the extracts caused only about a 7% loss in cell viability even at the highest concentration tested (2000 µg/mL), resulting in an IC_50_ between 458.2 and 638.1 µg/mL for skin cancer cells and between 827.5 and 1017 µg/mL for lung cancer cells. Regarding non-cancerous cell lines, after incubation with the leaf extracts, both HaCaT and HNTMC remained with more than 90% of viable cells, with IC_50_ values set between 1052 and 1106 µg/mL in the keratinocyte assay and between 309.6 and 998.3 µg/mL for fibroblasts.

Our results support previous studies in which therapeutic safety was demonstrated after exposure of BV-2, MSCs, and NIH-3T3 cells to extracts of guarana and metabolite caffeine, resulting in a low reduction in viability or even inducing cell proliferation [95,96,97]. On the other hand, previous research has shown antiproliferative effects on HL-60, HT-29, and MCF-7 cancer cells after incubation with guarana seed extract [73,98], and in another study the alkaloid theobromine, described in the phytochemical composition of *P. cupana*, was responsible for inhibiting the growth of the U87MG strain, causing obvious morphological changes [99].

### 3.8. Leishmanicidal Activity

The antileishmanial activity was evaluated in promastigote cultures of *L. amazonensis*, and the results are shown in Table 5 and Appendix A. The AgNPs-LD inhibited cell growth at concentrations of 100, 50, and 25 µg/mL, with a reduction of 76.03%, 49.40%, and 5.14%, respectively, resulting in a CI_50_ of 47.23 µg/mL. At these same concentrations, the AgNPs-LR showed cell death percentages of 84.06%, 52.91%, and 7.81%, with a CI_50_ of 43.79 µg/mL. In turn, the evaluation of the activity against *L. amazonensis* caused after incubation with AgNO_3_ resulted in a CI_50_ of 54.8 µg/mL, while the drug miltefosine was the most efficient in reducing the cell viability of the parasite, showing a CI_50_ of 17.25 µg/mL. Regarding the aqueous extracts, it was observed that after incubation with Ext-LD and Ext-LR at a concentration of 100 µg/mL, the inhibition was 7.82% and 22.76%, respectively, and from these values it was not possible to measure the IC_50_.

To assess toxicity against macrophages, which are the main host cells for *Leishmania* sp. parasites, the MTT test was carried out for 72 h, and the results are described in Table 5 and Appendix A. After exposure to AgNPs, the survival rate of RAW 264.7 macrophages remained close to 90% even at the highest concentrations. In addition, the aqueous extracts of *P. cupana* leaves showed viability of 96% at a concentration of 200 µg/mL. Based on these results, it was not possible to calculate the CC_50_ of the samples described above since higher concentrations would be needed to reduce cell viability, demonstrating the absence of cytotoxicity within the range of concentrations tested over the period evaluated.

On the other hand, exposure of macrophages to AgNO_3_ showed a reduction in cell viability from intermediate concentrations, with a CC_50_ of 61.35 µg/mL, suggesting greater toxicity than biogenic nanostructures. In addition, the action of miltefosine was evident, with a decrease in viability at the highest concentrations tested and a CC_50_ of 241.1 µg/mL. In short, as described and based on the SI values, the safety profile for the AgNPs tests was approximately twice as selective compared to incubation with the aqueous extracts and the metal salt, affecting the target cells to a lesser extent.

The information from this study corroborates other studies that have shown the leishmanicidal effect of biogenic AgNPs on the *L. amazonensis* species, even at varying concentrations and exposure times, without affecting macrophages [100,101]. Although extracts of *P. cupana* leaves did not have a significant effect on the loss of parasite viability, other studies have reported that plant extracts from Amazonian plants had an efficient antileishmanial action, with CI_50_ values between 5 and 20 µg/mL [102,103], as well as an absence of cytotoxicity for macrophages at concentrations of 10–400 µg/mL [104,105].

There are some suggested mechanisms of antileishmanial action caused by the size and surface charge of the nanostructures [106]; by the presence of phytochemicals used as coating agents, which can potentiate the activity through immunomodulatory effects [107]; and by the release of Ag^+^ ions when in contact with acidic organelles, such as the phagolysosome [108]. In addition, AgNPs can lead to a reduction in mitochondrial membrane potential, which maintains the vital energy of cells [109], causes an increase in the production of ROS and thus arrests cell cycle phases [110], increases the production of nitric oxide (NO) with the generation of microbicidal compounds [111], and inhibits the trypanothione/trypanothione reductase enzyme system [112].

### 3.9. Larvicidal and Pupicidal Activity Against Aedes aegypti

In this study, tests were carried out on third instar larvae of the *Ae. aegypti* mosquito after the organisms were exposed to AgNPs-LD and AgNO_3_. The results showed that the deleterious effects are time-dependent, with the LC_50_ and LC_90_—along with the lower and upper confidence limits—and R^2^ values shown in Table 6.

The dose-response curves were similar in terms of mortality rates (%), reaching lethality above 95% from a concentration of 5 µg/mL, with higher mortality rates as concentration and time increased (Appendix A). Overall, the larvae showed greater susceptibility to the nanostructures compared to the aqueous solution of the metal salt. The LC_50_ values for AgNPs-LD were 9.936 µg/mL (24 h), 1.669 µg/mL (48 h), and 0.6097 µg/mL (72 h), while for AgNO_3_ they were 9.179 µg/mL (24 h), 2.682 µg/mL (48 h), and 0.9730 µg/mL (72 h) (Table 6). These results corroborate other studies that have reported time-dependent effectiveness in eliminating *Ae. aegypti* larvae when exposed to biogenic AgNPs synthesized from extracts of different plant species, in which the LC_50_ after 24, 48, and 72 h was 9.2, 1.51, and 0.61 µg/mL, respectively [113,114,115]. In addition, an LC_50_ of 21.23 µg/mL was reported for aqueous AgNO_3_ solution after exposure to *Ae. aegypti* larvae for 24 h [116].

To gain a deeper understanding of the larvicidal potential of the samples, the LC_90_ values were calculated after carrying out the experiments. The concentrations reflecting the efficacy of the AgNPs were 15.64, 3.867, and 1.065 µg/mL after 24, 48, and 72 h, respectively. In turn, the LC_90_ of the AgNO_3_ solution was 14.5 µg/mL (24 h), 6.955 µg/mL (48 h), and 2.457 µg/mL (72 h) (Table 6). Thus, our results agree with previous studies that showed LC_90_ values of 12.11 and 13.96 µg/mL after exposure for 24 h to AgNPs synthesized from leaf extracts of *Derris trifoliata* and *Annona reticulata* [117,118].

Regarding the possible mechanisms of action, larvicidal activity can be attributed to the interaction of AgNPs with the extracellular lipoprotein matrix, causing an increase in the permeability of the plasma membrane of larval cells [119]. In addition, the generalist food habits and acidic environment of the insect midgut, as well as the generally negative zeta potential and the small diameter of the nanostructures, facilitate the internalization of AgNPs, mainly through the insect cuticle (external part of the exoskeleton) [120,121]. In the intracellular space, AgNPs bind to sulfur-containing proteins or phosphorus-containing compounds, such as DNA, leading to the denaturation of organelles and enzymes, as well as reducing ATP synthesis and ion exchange, which results in loss of function, morphological changes, and cell death [122,123].

In addition, the effects on pupae after exposure to AgNPs-LD and AgNO_3_ are described, showing the LC_50_ and LC_90_, as well as the confidence intervals and R^2^ values adjusted for each exposure time (Table 7).

In Appendix A, the dose-response curves show that the mortality rate at the end of the exposure period to the samples was 100% at the highest test concentrations and increased in a dose-dependent manner. As shown in Table 7, the greater the exposure time, the better the efficacy in eliminating pupae, with an LC_50_ of 6.885 µg/mL for AgNPs-LD after 24 h compared to AgNO_3_, which had 9.959 µg/mL. In turn, after 48 h, the LC_50_ was 2.840 and 4.194 µg/mL for the nanostructures and AgNO_3_, respectively. The LC_90_ values for the pupae were also defined based on the mortality rates after 24 and 48 h and reached, respectively, 11.98 µg/mL and 6.790 µg/mL for the AgNPs-LD and 14.82 µg/mL and 11.08 µg/mL for the metal salt.

Therefore, our results provide evidence that AgNPs synthesized with *P. cupana* leaf extract interfere in the *Ae. aegypti* life cycle and are in line with previous research in which biogenic AgNPs were tested against pupae of this mosquito, resulting in an LC_50_ of 13 µg/mL after 24 h of exposure [124,125]. In addition, delayed pupation and abnormal wing development of adult *Ae. aegypti* mosquitoes after incubation for 24 h with AgNPs synthesized from *Artemisia nilagirica* extract have already been reported [126].

In our study, we emphasized that the guarana leaf extract, when tested alone, did not show any activity against *Ae. aegypti* larvae, even at high concentrations (from 200 to 2000 µg/mL), and for this reason, this sample was not used in the pupicidal tests. It should be noted that the toxic mode of action of plant phytochemicals on insects can result in non-specific effects on a wide range of targets—including signaling molecules, ion channels, enzymes, proteins, nucleic acids, and biological membranes—that occur from the moment these secondary metabolites are ingested by herbivorous insects [127,128]. In addition, various polar biomolecules tend to dissolve in water and cause oxygen suppression, as well as penetrating the larvae via the respiratory tract, leading to lethal damage [129].

### 3.10. Catalytic Degradation of Organic Dyes

The catalytic degradation efficiency of AgNPs was investigated in the reduction of the cationic dye methylene blue (MB), and as shown in Figure 9A, the UV-Vis spectra of the dye alone over 90 min show no decrease in the absorption intensity of its characteristic bands around 290 and 664 nm, with a pronounced shoulder at 614 nm. On the other hand, the degradation process is accelerated with the addition of AgNPs, and after 40 min of reaction, it is possible to observe the disappearance of the blue color in the dye solutions regardless of the type of nanocatalyst used (Figure 9B), as well as a decay of the MB absorption bands in the UV-Vis spectra (Figure 9C,D). As the intense band around 664 nm decreases, a new intense band appears near 256 nm, which can be attributed to the formation of leucomethylene blue [32].

As shown in Appendix A, the percentage elimination of MB was 91.12% by AgNPs-LD and 93.52% by AgNPs-LR after 40 min of reaction. In addition, we found that the model best suited to the degradation of the pollutant followed second-order kinetics for AgNPs-LD, with an R^2^ of 0.9762 and a reaction rate constant (*k*) of 0.1218 min^−1^ (Appendix A), while for AgNPs-LR, first-order kinetics was more suitable, with R^2^ and rate constant values of 0.945 and 0.0579 min^−1^, respectively (Appendix A). Our findings are like recent reports in which biogenic nanocatalysts produced from *Carissa opaca* and *Ficus sycomorus* extracts promoted the complete reduction of MB after 50 min, with a rate constant of 0.031 min^−1^ and 12 min with a *k* value of 0.156 min^−1^, respectively [130,131].

The catalytic activity of AgNPs against the anionic dye methyl orange (MO) was also investigated, and as shown in Figure 10A, in the absence of the nanocatalysts, there is no decrease in absorbance intensity in the spectral bands characteristic of the organic molecule at 270 and 464 nm after 60 min. Successful degradation of the dye is achieved after 14 min of reaction in the presence of AgNPs, which considerably reduce the intensity of the color in the solutions (Figure 10B). In addition, the absorption band at 464 nm decays, and the band at 270 nm disappears, while a new band at 247 nm appears due to absorptions of the -NH2 group from hydrazine molecules, sulphanilic acid, and aromatic amines that are products of MO degradation (Figure 10C,D) [132].

The percentage of MO degradation is time-dependent, and at the end of 14 min, it was 92.89% and 96.42% for AgNPs-LD and AgNPs-LR, respectively (Appendix A). When studying the models best suited to the chemical kinetics of each reaction, it was found that the zero-order model resulted in the best characteristics after incubation with both types of nanostructures, with R^2^ values of 0.9176 and 0.9498 for AgNPs-LD and AgNPs-LR, as well as reaction rate constant (*k*) values of 0.0945 min^−1^ and 0.0946 min^−1^, respectively (Appendix A). Recently, some studies have investigated the catalytic potential of AgNPs synthesized from extracts of *Berberis vulgaris* and *Heterotheca subaxillaris*, which resulted in complete degradation of the MO dye between 9 and 11 min, as well as reaction rate constants between 0.120 and 0.4109 min^−1^; the higher these values, the faster the degradation of the pollutants [133,134].

The catalytic properties of metal nanoparticles depend on their redox potential, which must be higher than the reducing potential of the substrate and lower than that of the reactants [135]. According to our results, even though NaBH_4_ alone is a strong reducing agent, it is not able to reduce MB and MO, and this is due to the considerable difference in redox potential between them, which makes the reactions thermodynamically permissible but kinetically non-advantageous [136]. The accelerated rate of reduction of organic dyes in the presence of AgNPs can be attributed to the small diameter of these nanostructures and, consequently, their large surface area, which offers more active sites for the reactions to take place [137,138].

Catalytic activity can be studied based on the Langmuir–Hinshelwood (L-H) model, in which AgNPs act through the ‘electron relay’ effect [139,140]. Thus, the degradation of organic pollutants is based on an electron transfer process that occurs after the nanocatalysts are added to the reaction mixture, with subsequent adsorption of the dissociated NaBH_4_ molecules (nucleophiles) on their surface by means of electrostatic attraction. The AgNPs then act as mediators in the sharing of electrons for the dye molecules (electrophiles), which causes modifications to the chemical structures of the pollutants; the reduction products formed are stable and non-toxic, and when they are desorbed from the surface of the AgNPs, they melt into the solution, which becomes colorless [141,142,143].

## 4. Conclusions

The extracts of *P. cupana* leaves collected at different times of the year were efficient at reducing silver ions and stabilizing AgNPs, highlighting the importance of assessing this parameter to optimize the yield and final characteristics of the nanostructures formed using a plant native to Brazilian biodiversity with economic and social value.

The AgNPs showed similar characteristics, including high colloidal stability under the storage conditions evaluated, as well as varied morphologies and small diameters, which may have contributed to the biological activities investigated. Significant activities were observed in reducing bacterial growth of Gram-positive and Gram-negative species, as well as non-specific cytotoxic activity against cancerous and non-cancerous cells, in addition to significant elimination of free radicals in a dose-dependent manner, like that observed for plant extracts. The AgNPs showed insecticidal potential, affecting the development of disease-carrying mosquitoes, as well as marked activity against opportunistic parasites of human host cells and promising catalytic degradation rates of organic dyes at room temperature in short reaction times.

These findings broaden knowledge about the green synthesis of AgNPs, contributing to the continuity of studies related to the conditions in which plants are used to produce nanomaterials and reinforcing the need to promote the sustainable use of natural resources with a view to increasing biomedical and environmental applications.

## Figures and Tables

**Figure 1 pharmaceutics-17-00356-f001:**
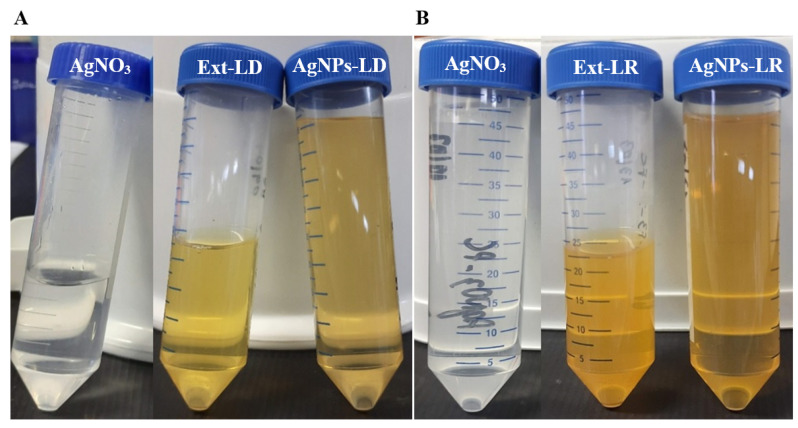
Visual recording of the 2 mM silver nitrate solutions, plant extracts from *Paullinia cupana* leaves, and biogenic AgNPs.

**Figure 2 pharmaceutics-17-00356-f002:**
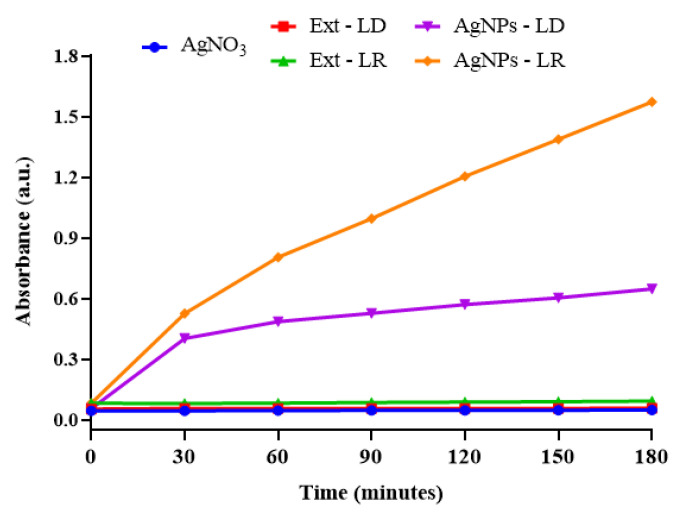
Kinetic curves for the synthesis of AgNPs and their experimental controls obtained by spectrophotometric analysis at 450 nm for 180 min of reaction.

**Figure 3 pharmaceutics-17-00356-f003:**
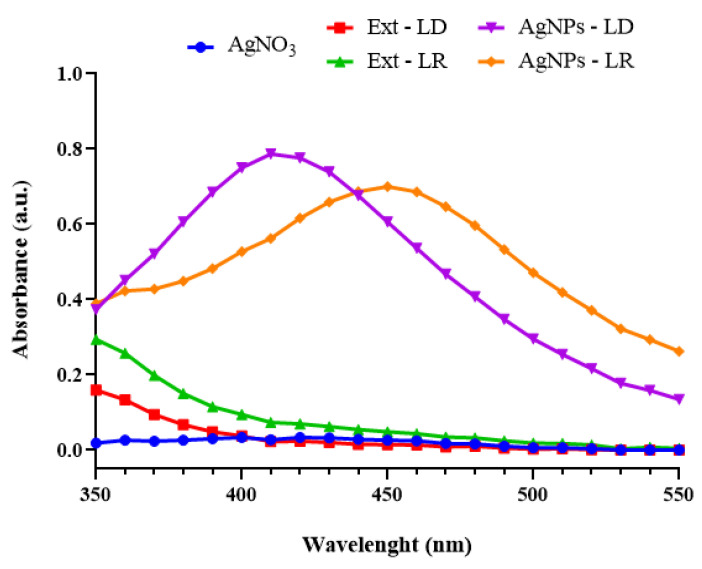
Absorption curves of AgNPs and their experimental controls measured by spectrophotometry in the 350–550 nm range after 180 min of reaction.

**Figure 4 pharmaceutics-17-00356-f004:**
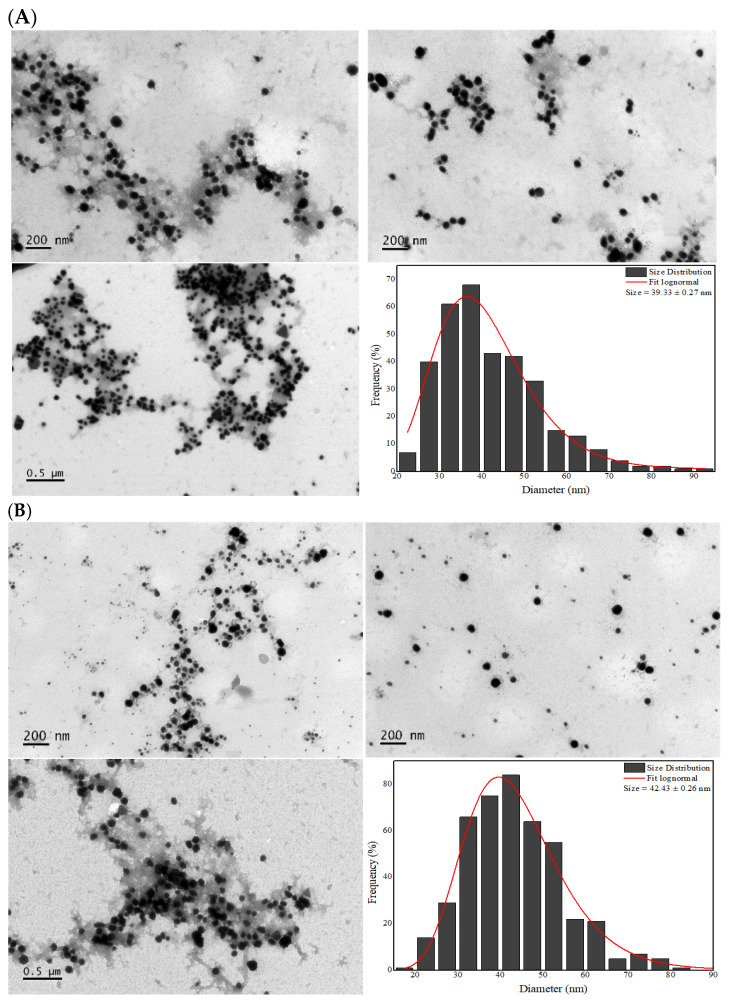
TEM micrographs at 10,000× magnification and histograms of the dry diameter distribution of (**A**) AgNPs-LD and (**B**) AgNPs-LR.

**Figure 5 pharmaceutics-17-00356-f005:**
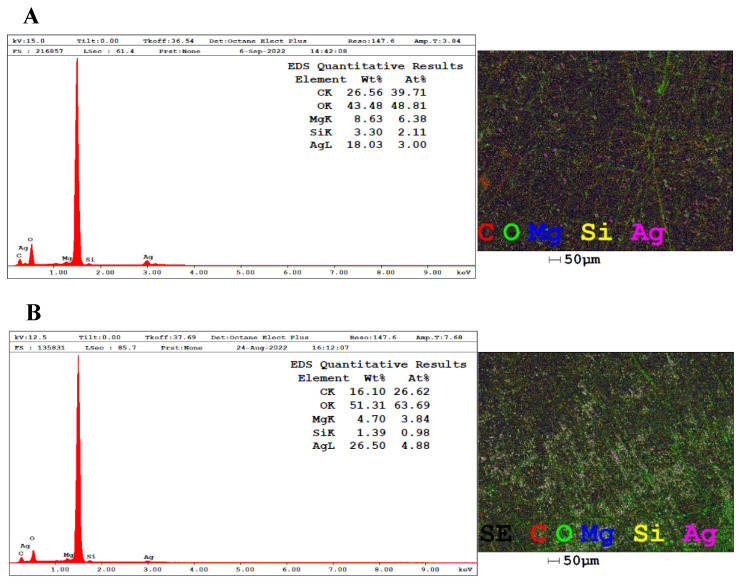
EDX spectra and elemental map showing the distribution of the atoms present in the samples (**A**) AgNPs-LD and (**B**) AgNPs-LR.

**Figure 6 pharmaceutics-17-00356-f006:**
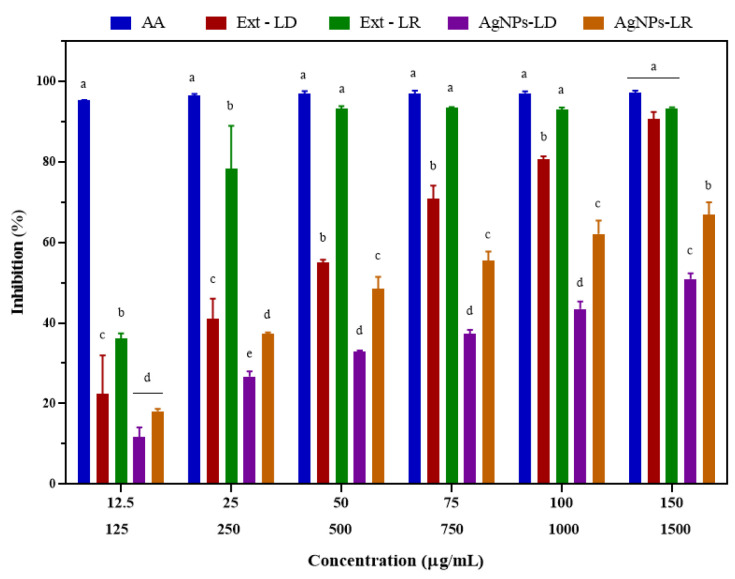
Antioxidant activity of AgNPs, plant extracts of *Paullinia cupana* leaves collected in the dry and rainy seasons, and aqueous ascorbic acid (AA) solution against the DPPH free radical. The different letters indicate statistical significance (*p* < 0.05) between the treatments within each test concentration. On the X axis, the upper line indicates the test concentrations of AgNPs and AA. The lower line indicates the test concentrations of the plant extracts of *Paullinia cupana* leaves.

**Figure 7 pharmaceutics-17-00356-f007:**
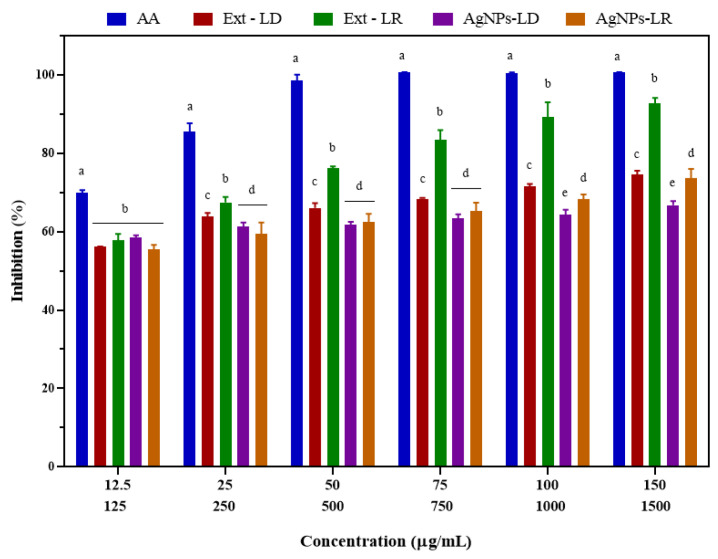
Antioxidant activity of AgNPs, plant extracts of *Paullinia cupana* leaves collected in the dry and rainy seasons, and aqueous ascorbic acid (AA) solution against the ABTS free radical. The different letters indicate statistical significance (*p* < 0.05) between the treatments within each test concentration. On the X axis, the upper line indicates the test concentrations of AgNPs and AA. The lower line indicates the test concentrations of the plant extracts of *Paullinia cupana* leaves.

**Figure 8 pharmaceutics-17-00356-f008:**
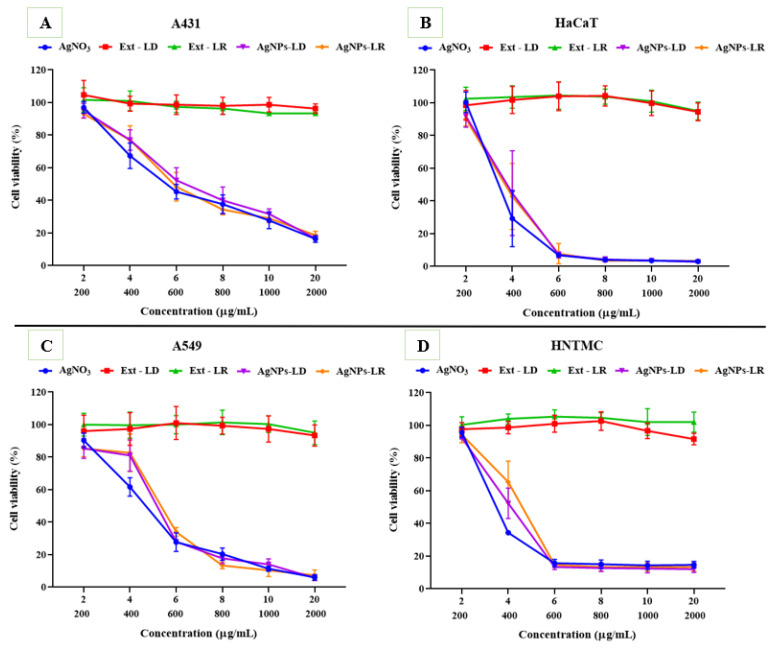
Viability graphs of the cell lines (**A**) A431, (**B**) HaCaT, (**C**) A549, and (**D**) HNTMC after incubation with AgNPs, plant extracts of *Paullinia cupana* leaves collected in the dry and rainy seasons, and aqueous AgNO_3_ solution. On the X axis, the top line indicates the test concentrations of AgNPs and AgNO_3_. The lower line indicates the test concentrations of the *Paullinia cupana* leaf extracts.

**Figure 9 pharmaceutics-17-00356-f009:**
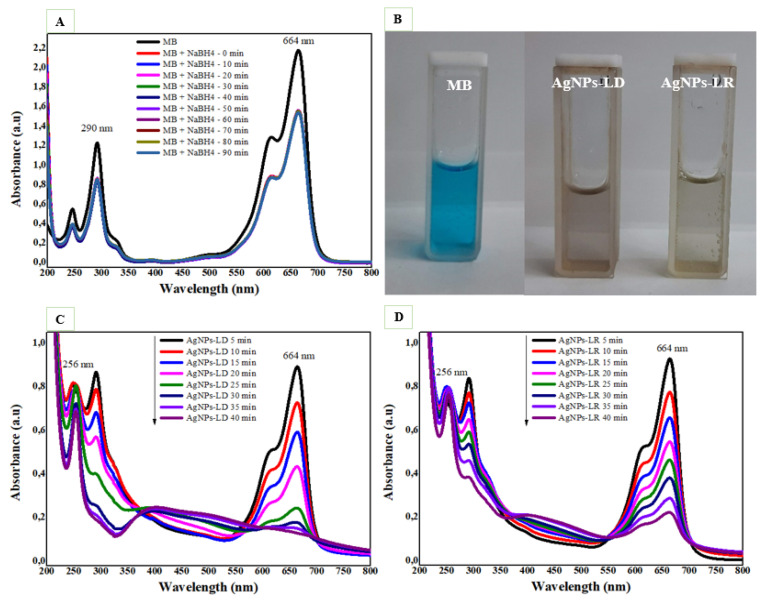
Catalytic degradation of methylene blue dye (MB): (**A**) UV-Vis absorption spectra with the NaBH_4_ substrate only; (**B**) images of aqueous solutions of MB in the absence and presence of AgNPs (**C**) UV-Vis spectra with NaBH_4_ and the AgNPs-LD; and (**D**) AgNPs-LR nanocatalysts.

**Figure 10 pharmaceutics-17-00356-f010:**
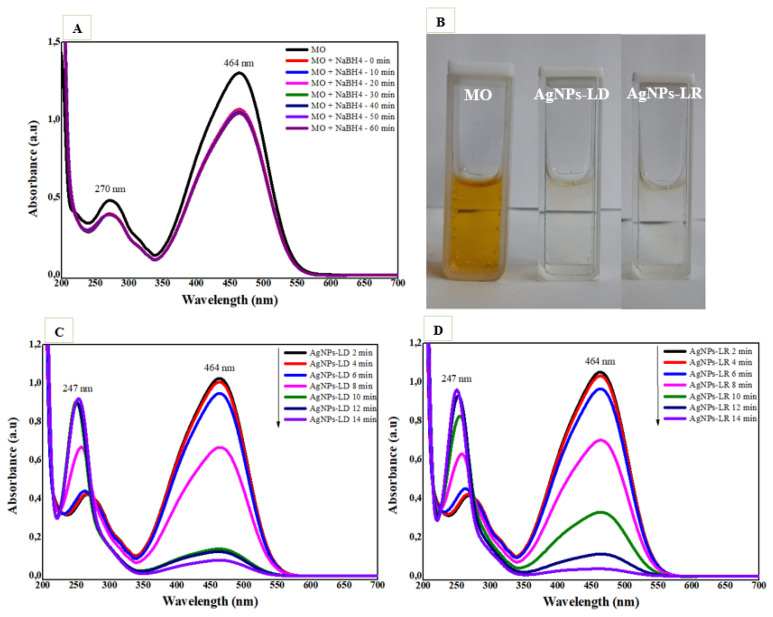
Catalytic degradation of methyl orange dye (MO): (**A**) UV-Vis absorption spectra with the NaBH_4_ substrate only; (**B**) images of aqueous solutions of MO in the absence and presence of AgNPs (**C**) UV-Vis spectra with NaBH_4_ and the AgNPs-LD; and (**D**) AgNPs-LR nanocatalysts.

**Table 1 pharmaceutics-17-00356-t001:** Monitoring the colloidal stability of AgNPs-LD by means of dynamic light scattering (DLS) and surface zeta potential analyses over two years with storage at room temperature (RT) or in a refrigerator (REF).

Time/Storage	AgNPs-LD		
	HD (nm)	PdI	ZP (mV)
D0	81.90 ± 5.5	0.448 ± 0.120	−38.1 ± 0.5
D1-RT	57.54 ± 8.3	0.361 ± 0.121	−36.9 ± 2.0
D1-REF	71.20 ± 20.5	0.354 ± 0.228	−22.4 ± 11.5 *
D7-RT	54.04 ± 1.3 *	0.318 ± 0.046	−32.3 ± 1.5
D7-REF	67.20 ± 14.9	0.450 ± 0.055	−29.5 ± 2.2
D14-RT	53.34 ± 1.7 *	0.408 ± 0.012	−32.3 ± 3.6
D14-REF	76.52 ± 17.7	0.402 ± 0.109	−32.9 ± 7.6
D40-RT	56.58 ± 1.1	0.388 ± 0.035	−35.8 ± 0.7
D40-REF	65.07 ± 11.1	0.358 ± 0.105	−35.3 ± 1.1
D60-RT	56.96 ± 0.4	0.406 ± 0.008	−15.1 ± 0.8 *
D60-REF	59.27 ± 6.3	0.322 ± 0.121	−30.5 ± 1.4
D90-RT	60.30 ± 1.4	0.359 ± 0.036	−22.7 ± 1.0 *
D90-REF	56.81 ± 6.2	0.379 ± 0.133	−22.6 ± 0.7 *
D180-RT	68.33 ± 0.3	0.302 ± 0.008	−29.2 ± 0.8
D180-REF	61.59 ± 1.9	0.455 ± 0.009	−28.0 ± 2.0
D365-RT	65.84 ± 0.5	0.361 ± 0.006	−26.0 ± 3.9
D365-REF	64.26 ± 2.0	0.416 ± 0.002	−25.3 ± 5.9 *
D730-REF	63.08 ± 0.3	0.411 ± 0.019	−27.8 ± 0.6

Values are represented as the mean ± standard deviation of the mean of measurements in triplicate. Statistical analysis: one-way ANOVA test (*p* < 0.05) followed by Tukey’s test. The superscript symbol (*) indicates significant differences within each analysis parameter for the AgNPs-LD compared to the measurement on day 0 (D0).

**Table 2 pharmaceutics-17-00356-t002:** Monitoring the colloidal stability of AgNPs-LR by means of dynamic light scattering (DLS) and surface zeta potential analyses over two years with storage at room temperature (RT) or in a refrigerator (REF).

Time/Storage	AgNPs-LR		
	HD (nm)	PdI	ZP (mV)
D0	91.74 ± 1.1	0.295 ± 0.019	−34.9 ± 1.4
D1-RT	85.37 ± 3.1	0.298 ± 0.041	−37.2 ± 6.2
D1-REF	89.68 ± 3.9	0.330 ± 0.010	−34.0 ± 0.7
D7-RT	88.38 ± 1.4	0.278 ± 0.058	−29.3 ± 0.6 *
D7-REF	87.85 ± 1.3	0.273 ± 0.012	−29.5 ± 0.6 *
D14-RT	89.10 ± 0.3	0.230 ± 0.012	−31.6 ± 1.0
D14-REF	92.27 ± 1.3	0.276 ± 0.017	−31.4 ± 0.5
D40-RT	89.94 ± 0.8	0.227 ± 0.015	−33.3 ± 0.2
D40-REF	94.35 ± 2.2	0.241 ± 0.076	−34.5 ± 0.2
D60-RT	88.80 ± 1.1	0.225 ± 0.013	−34.9 ± 0.7
D60-REF	94.75 ± 3.2	0.214 ± 0.026	−34.3 ± 0.7
D90-RT	89.59 ± 0.9	0.244 ± 0.019	−33.5 ± 0.4
D90-REF	96.11 ± 3.1	0.256 ± 0.015	−34.3 ± 0.5
D180-RT	89.77 ± 0.8	0.261 ± 0.007	−31.1 ± 1.0
D180-REF	100.90 ± 1.6 *	0.278 ± 0.021	−31.5 ± 0.3
D365-RT	90.83 ± 2.2	0.321 ± 0.041	−23.5 ± 1.7 *
D365-REF	90.18 ± 1.3	0.287 ± 0.019	−29.4 ± 0.1 *
D730-REF	126.2 ± 2.9 *	0.362 ± 0.014	−27.1 ± 0.3 *

Values are represented as the mean ± standard deviation of the mean of measurements in triplicate. Statistical analysis: one-way ANOVA test (*p* < 0.05) followed by Tukey’s test. The superscript symbol (*) indicates significant differences within each analysis parameter for the AgNPs-LD compared to the measurement on day 0 (D0).

**Table 3 pharmaceutics-17-00356-t003:** Characteristics of the biosynthesized AgNPs obtained by nanoparticle tracking analysis.

Samples	AgNPs-LD	AgNPs-LR
Size (nm)	68.5 ± 0.7	89.3 ± 2.1
Mode (nm)	62.3 ± 1.7	73.1 ± 1.6
Concentration (particles/mL)	5.33 × 10^7^ ± 3.04 × 10^6^	1.5 × 10^8^ ± 1.09 × 10^7^
*Span* index	0.443	0.523

The values are represented as the mean ± standard deviation of the mean of triplicate measurements.

**Table 4 pharmaceutics-17-00356-t004:** Minimum inhibitory concentration (MIC) and minimum bactericidal concentration (MBC) of AgNPs and AgNO_3_ aqueous solution on various strains of bacteria.

Microorganisms	AgNPs-LD		AgNPs-LR		AgNO_3_	
	MIC	MBC	MIC	MBC	MIC	MBC
*A. baumannii*	21.25	21.25	10.60	21.25	5.30	5.30
*B. cereus*	21.25	42.50	10.60	42.50	5.30	5.30
*E. coli*	21.25	21.25	21.25	21.25	10.60	21.25
*K. pneumoniae*	21.25	21.25	21.25	21.25	10.60	10.60
*P. aeruginosa*	21.25	42.50	21.25	42.50	21.25	42.50
*S. enterica*	42.50	42.50	21.25	42.50	10.60	10.60
*S. aureus*	21.25	21.25	21.25	21.25	10.60	10.60
*S. epidermidis*	21.25	42.50	21.25	42.50	10.60	10.60

MIC and MBC = µg/mL.

**Table 5 pharmaceutics-17-00356-t005:** Antileishmanial activity (CI_50_), cytotoxicity against macrophages (CC_50_), and selectivity index (SI) values calculated after exposures for 72 h with AgNPs, plant extracts of *Paullinia cupana* leaves, aqueous AgNO_3_ solution, and the drug miltefosine.

Samples	Macrophages	Promastigotes	SI
	CC_50_	CI_50_	
AgNPs-LD	>100	47.23	>2.12
AgNPs-LR	>100	43.79	>2.28
Ext-LD	>100	>100	>1
Ext-LR	>100	>100	>1
AgNO_3_	61.35	54.8	1.12
Miltefosine	241.1	17.25	13.97

CC_50_ and CI_50_ = µg/mL; SI = CC_50_/CI_50._

**Table 6 pharmaceutics-17-00356-t006:** Insecticidal activity of AgNPs-LD and aqueous AgNO_3_ solution against third instar *Aedes aegypti* larvae at 24, 48, and 72 h of exposure.

Sample	Time (Hours)	LC_50_ (µg/mL) (CI 95%)	LC_90_ (µg/mL) (CI 95%)	R^2^
AgNPs-LD	24	9.936 (9.273–10.22)	15.64 (13.96–17.23)	0.8451 (LC_50_) 0.8434 (LC_90_)
	48	1.669 (1.347–2.040)	3.867 (3.137–4.571)	0.8470 (LC_50_) 0.8313 (LC_90_)
	72	0.6097 (0.5475–0.6849)	1.065 (0.8841–1.240)	0.9322 (LC_50_) 0.9266 (LC_90_)
AgNO_3_	24	9.179 (8.345–9.873)	14.50 (12.99–16.03)	0.8862 (LC_50_) 0.8822 (LC_90_)
	48	2.682 (2.335–3.071)	6.955 (5.695–8.100)	0.9242 (LC_50_) 0.9231 (LC_90_)
	72	0.9730 (0.8478–1.114)	2.457 (2.196–2.701)	0.9452 (LC_50_) 0.9373 (LC_90_)

LC_50_: lethal concentration for 50% of the larvae; LC_90_: lethal concentration for 90% of the larvae; CI: confidence interval (lower/upper); R^2^: correlation coefficient.

**Table 7 pharmaceutics-17-00356-t007:** Insecticidal activity of AgNPs-LD and aqueous AgNO_3_ solution against *Aedes aegypti* pupae at 24 and 48 h of exposure.

Sample	Time (Hours)	LC_50_ (µg/mL) (CI 95%)	LC_90_ (µg/mL) (CI 95%)	R^2^
AgNPs-LD	24	6.885 (6.172–7.684)	11.98 (10.42–13.38)	0.7822 (LC_50_) 0.7854 (LC_90_)
	48	2.840 (2.139–3.563)	6.790 (5.279–8.275)	0.6393 (LC_50_) 0.6391 (LC_90_)
AgNO_3_	24	9.959 (9.164–10.70)	14.82 (13.42–16.01)	0.9134 (LC_50_) 0.9194 (LC_90_)
	48	4.194 (3.305–5.187)	11.08 (8.614–13.50)	0.6240 (LC_50_) 0.6344 (LC_90_)

LC_50_: lethal concentration for 50% of the larvae; LC_90_: lethal concentration for 90% of the larvae; CI: confidence interval (lower/upper); R^2^: correlation coefficient.

## Data Availability

The data presented in this study are available on request from the authors.

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
