# Peer review of "Green Synthesis of Silver Nanoparticles Using Paullinia cupana Kunth Leaf Extract Collected in Different Seasons: Biological Studies and Catalytic Properties"

_pharmaceutics, 2025, doi:10.3390/pharmaceutics17030356_

Round 1

Reviewer 1 Report

Comments and Suggestions for Authors

A large-scale study of the physicochemical, chemical and biological activities of silver nanoparticles obtained by the interaction of silver nitrate and extracts of guarana collected in different seasons has been carried out. This comparison makes sense, since environmental factors determine the changes in the composition of secondary metabolites in plants, and hence their potential activity. The resulting particles are well characterized in terms of size, colloidal stability and other physicochemical factors, but there is no clarity regarding the composition of the extracts used and the seasonal differences that occurred in the secondary metabolites.

The antibacterial, antioxidant, insecticidal and antiparasitic activity against leishmaniasis has been studied, but not all studies are well motivated and logically connected. 

In fact, the article can be divided into two smaller ones, better traceable as a relationship, as well as easier for the reader to perceive and analyze.

The formula for calculating the IC50 of plant extracts in the study of cytotoxicity is not specified, given that the survival rate remains in high limits even at the highest tested concentration of 2 mg/ml, and the IC50 is specified as close to 1 mg/ml, where the survival rate is over 90%. How was such a value established for plant extracts?

The statistical significance of the observed differences in the activity of the extracts collected in two different seasons was not assessed and not discussed. There are also no indications of differences in the composition of the extracts that could explain the observed differences in activity and physicochemical characteristics (e.g. size). There is no statistical evaluation of the results in Table 5, 6, 7 and Figure 8. 

From the title and introduction, the reader is left with the impression that the emphasis will be placed precisely on the differences in the composition and activity of the plant's secondary metabolites as a result of the different natural conditions in the two seasons and that there will be a comparative analysis, but such an analysis was not made.
How do you explain the lack of insecticidal effect of the extracts collected in the rainy season, when the conditions for mosquito reproduction are optimal? What is the basis for comparing plant extracts and nanoparticles with a tenfold difference in concentrations in terms of antioxidant activity? The comparison should be made at concentrations of the extracts close to those in the nanoparticles. The obvious differences in the activity of the extracts collected in the two seasons are not discussed.
Some of the formulas and designations are not in English (for example, formula 4).

Author Response

Dear reviewer,
The answers from us, the authors, are attached.
Thank you very much

Reviewer 2 Report

Comments and Suggestions for Authors

I congratulate the authors for a comprehensive and complex work.
General observations:
- line 131 - the name of the microorganisms in italics
- line 395 in ec 4 written in English or symbol
- tables 4, 6,7,8 do not present the standard deviations, considering that a statistical analysis of the data performed in triplicate was specified, and in the supplementary material these deviations are shown in the graphs
- figures 6,7 and 8 have both AgNPs and extract concentration on the abscissa. The authors should decide on the important component and refer to it as its concentration.

Comments on the Quality of English Language

Pay attention to the writing format of the names of microorganisms!

Author Response

(The authors gave the same response as above.)

Reviewer 3 Report

Comments and Suggestions for Authors

The manuscript submitted to Pharamceutics entitled “Green synthesis of silver nanoparticles using Paullinia cupana Kunth leaf extract collected in different seasonal: biological 3 studies and catalytic properties” describes various potentials of the synthesized silver nanoparticles (AgNPs) synthesized utilizing seasonal extract of guarana leaves, an Amazonian shrub listed in the Brazilian Pharmacopoeia. The methods utilized traditional knowledge to develop the nanotechnology for the generation of silver nanoparticles might generates less toxic waste in the process. The synthesized silver nanoparticles (AgNPs) are effective against the Aedes 53 aegypti larvae and pupae is a high point along with other known properties that the authors have tested and found the success. Overall, the manuscript is interesting, well-written and shows the effectiveness of traditional and advance knowledge. Authors did the study for years. Procedures and experiments are well designed, and sufficient information is provided. All the figure legends and information are accurately provided. This is a vast study compiled in the manuscript, and I would like to recommend the manuscript for publication with minor revision.

High pints of the manuscript.

  1. Authors have synthesized silver nanoparticles via green synthesis
  2. Provided every possible experimental detail
  3. Synthesized silver nanoparticles showed high colloidal stability
  4. Investigated various biological activities
  5. Catalytic degradation of methylene blue (MB) dye

Kudos to authors, I was a great read!

Comments:

  1. What would contribute to such biological activities of the AgNPs? Could it be the diameter of the nanoparticle or stability?
  2. Is there a role extracts of leaves of Cupana, (which authors claimed) then why not call it synergy?
  3. Do the authors know any types of natural products presents in the extract that may contribute to the properties of AgNPs and biactivities; if yes, could the authors please name a few of them.

Author Response

(The authors gave the same response as above.)

Round 2

Reviewer 1 Report

Comments and Suggestions for Authors

Similar results, referring to the physicochemical parameters of the studied nanoparticles, have already been published in the previous publication indicated by the authors. This makes it unnecessary to repeat these results (preparation, hydrodynamic stability, dimensions and elemental composition) without introducing new data. Some of the studies on biological activity are also repeated (antibacterial and antioxidant) and thus have only a confirmatory nature, unnecessarily burdening and verifying the volume of the article.
The calculation of IC50 can be done in two ways: relative and absolute. The relative EC50/IC50 is the concentration corresponding to a response midway between the estimates of the lower and upper plateaus. The absolute EC50/IC50 is the response corresponding to the 50% control (the mean of the 0% and 100% assay controls). In the case of cytotoxicity studies, the absolute value is applicable, not the relative one, which the authors probably applied and which leads to erroneous conclusions. In view of this, it is advisable to recalculate the IC50 values ​​compared to the absolute ones. I recommend that the article be edited with an emphasis on the new results, a discussion of the possible reasons for the observed differences in effect, and a reference to the already published results on nanoparticles with plant extracts from flowers and leaves collected in dry and rainy periods.

Author Response

Dear reviewer
Our responses to your comments are in the attached document.
We strongly hope that you will understand what is described so that we can continue with the submission process.
Thank you!

Round 3

Reviewer 1 Report

Comments and Suggestions for Authors

You are using appropriate software, but the parameters set are important for the accuracy of the analysis. Since you insist on publishing these data, I will agree to this. However, I think it is not correct to claim that only 50% of the cells survive at a concentration of about 1 mg/ml, since over 95% are viable in all treatments. Good luck!